# Fostering Prevention of Cervical Cancer by a Correct Diagnosis of Precursors: A Structured Case-Based Colposcopy Course in Finland, Norway and UK

**DOI:** 10.3390/cancers12113201

**Published:** 2020-10-30

**Authors:** Forsell Sabrina, Kalliala Ilkka, Halttunen-Nieminen Mervi, Redman Charles, Leeson Simon, Tropé Ameli, Moss Esther, Kyrgiou Maria, Pyörälä Eeva, Nieminen Pekka

**Affiliations:** 1Department of Obstetrics and Gynaecology, Helsinki University Hospital and Helsinki University, 00290 Helsinki, Finland; ilkka.kalliala@hus.fi (K.I.); mervi.halttunen-nieminen@hus.fi (H.-N.M.); 2Department of Metabolism, Digestion and Reproduction & Department of Surgery and Cancer, Faculty of Medicine, Imperial College London, Department of Surgery and Cancer, IRDB, Faculty of Medicine, Imperial College, London W12 0NN, UK; m.kyrgiou@imperial.ac.uk; 3Department of Obstetrics and Gynaecology, Royal Stoke University Hospital, Stoke-on-Trent ST4 6QG, UK; cweraf@icloud.com; 4Department of Obstetrics and Gynaecology, Ysbyty Gwynedd, Bangor, Gwynedd LL57 2PW, UK; simon.leeson@wales.nhs.uk; 5Norwegian Cervical Cancer Programme, Cancer Registry of Norway, NO-0304 Oslo, Norway; ameli.trope@mac.com; 6Department of Gynaecological Oncology, University Hospitals of Leicester, Leicester LE2 7LX, UK; em321@le.ac.uk; 7West London Gynaecological Cancer Centre, Imperial Healthcare NHS Trust, London W2 1NY, UK; 8Centre for University Teaching and Learning, University of Helsinki, 00290 Helsinki, Finland; eeva.pyorala@helsinki.fi

**Keywords:** colposcopy, colposcopic diagnosis, learning, CIN, HPV, cervical intra-epithelial neoplasia, case-based learning

## Abstract

**Simple Summary:**

Cervical cancer prevention is presently undergoing a thorough reformation due to introduction of human papillomavirus (HPV)-testing and vaccines in primary prevention. The screening program, however, is more than a single test or preventive intervention—the possible lesion has to be found, located and treated. Colposcopy plays a major role in this management. Literature dealing with training and learning, especially with colposcopy, is extremely scarce. The aim of the European Federation of Colposcopy, EFC, is to improve the education and training in colposcopy, e.g., by organizing colposcopy courses. The aim of our prospective interventional study was to pilot this intensive participant activating EFC Basic Colposcopy Course on the short- and long-term learning of colposcopy-related knowledge, image recognition and the diagnostic confidence.

**Abstract:**

High-quality colposcopy is essential in cervical cancer prevention. We performed a multicentre prospective interventional pilot-study, evaluating the effect of a six-hour case-based colposcopy course on short- and long-term learning of colposcopy-related knowledge, diagnostic accuracy levels and confidence. We recruited 213 colposcopists participating in three European Federation of Colposcopy (EFC) basic colposcopy courses (Finland, Norway, UK). The study consisted of three tests with identical content performed before, after and 2 months after the course, including ten colposcopic images, ten patient cases and scales for marking confidence in the answers. Outcome measures where mean scores in correct case-management, diagnosis (including high-grade lesion recognition), transformation-zone recognition and confidence in answers. Results were compared between the three tests and stratified according to experience. Mean test scores improved after the course for all participants. The increase was highest for beginners. Confidence in answers improved and the number of colposcopists showing high confidence with low scores decreased. A structured case-based course improves skills and confidence especially for inexperienced colposcopists; however, trainers should be aware of the risk of overconfidence. To complement theoretical training, further hands-on training including high-quality feedback is recommended. Conclusions drawn from long-term learning are limited due to the low participation in the follow-up test.

## 1. Introduction

Colposcopy is an essential step in the prevention of cervical cancer. The European Federation for Colposcopy (EFC) has developed a core curriculum for the necessary skills and knowledge required for those that wish to practice colposcopy independently [1]. In the UK, the BSCCP (British Society for Colposcopy and Cervical Pathology) and, in Finland, the SKY (Finnish Colposcopy Society) have developed a structured training program for colposcopy that is very similar in the two countries.

Although the required skills have been defined, there are no standardized methods for teaching and assessing skills improvement [2,3,4,5,6]. Only few studies have been performed on different methods for learning colposcopy [5,7,8,9]. Zahid and colleagues [9] demonstrated a significant improvement in knowledge of participants in the EFC basic colposcopy course in 2012. This questionnaire-based survey across three countries (Sweden, Germany and Belgium) focused on management of cases and produced heterogenous and inconsistent results across locations. The overall result showed an increase in case management performance after the course, but the change was significant only in one of the locations (Stockholm, Sweden). Data on the participants’ experience of colposcopies was not collected, so variations in results may reflect their differences in experiences of colposcopies. Despite theoretical knowledge of the principles underpinning management of cervical pre-invasive disease being important, this is only one aspect of colposcopy training, while mastering the ability to interpret visual findings is fundamental. Several studies have demonstrated inter- and intra-observer variability in colposcopy in both diagnosis and TZ-type (transition zone type) determination, indicating a need for high quality training [4,10,11].

EFC has developed a one-day case-based colposcopy course that has been offered for many years. The course is designed primarily for trainee colposcopists, but a small number of experienced colposcopists have attended these courses to update their skills. Active learning during the course is promoted by using patient cases and diagnostic images as triggers for learning and encouraging the course participants to actively solve clinical problems. In this study, the same structured course was delivered in Finland, Norway and the United Kingdom. The aim of our study was to evaluate whether this standardized basic case-based colposcopy course improved the ability to manage colposcopy patient cases, diagnostic accuracy and confidence [12]. Many studies have shown that colposcopy has a limited sensitivity for high-grade lesions [13,14,15,16,17]. In this study, we also evaluated the effect of the course on recognition of high-grade (CIN2+ (cervical intraepithelial neoplasia)) vs. low-grade lesion and normal vs. abnormal lesions. The evaluation was done in the short- and long-term and compared between colposcopists with different levels of experience in performing colposcopies.

## 2. Results

Altogether 213 colposcopists took part in this study (70 in Finland, 40 in Norway, 103 in the UK) (Table 1, Figure 1). The mean score in the case questions increased immediately after the course compared to before (*p* < 0.01 for pre- to post-course comparison) and, two months later, remained higher than the score before the course (*p* < 0.01) (Figure 2a). In image recognition correct diagnosis (*p* = 0.03) and TZ-type detection (*p* < 0.01) were higher immediately after the course compared to the scores obtained before the course. In the subcategories (high- vs. low-grade lesion and normal vs. abnormal lesion recognition), there was no improvement after the course, but the detection of abnormal lesion was better in the follow-up test compared to the pre-course test (*p* = 0.045) (Figure 2b). 

Among novice colposcopists (beginners), the greatest improvement in case-question scores was observed immediately after the course compared to their pre-course results (*p* < 0.001). Beginners scored as high after the course as experts before the course. For intermediate and expert colposcopists, scores increased in the case-question section (*p* < 0.001 and *p* = 0.02), but less than for the beginners. In the image questions, TZ type identification scores increased for beginners (*p* = 0.02) and intermediates (*p* = 0.004), but results indicating a correct diagnosis (*p* = 0.04) improved significantly only for beginners (Table 2). The confidence scores in the pre-course assessment were higher in the case part of the test than in the image part (Figure 2c). In both parts of the pre-course test, the levels of confidence rates were the lowest among beginners (62.1 mm (95% CI; 56.6–67.6)) and highest among the expert colposcopists. (86.5 mm (95% CI; 82.6–90.4)). The same was observed in the image part of the test (mean value for confidence among beginners was 35.1 mm (95% CI; 30.1–40.1) and 61.1 mm (95% CI; 56.4–65.9) for experts).

Compared to the pre-test, the mean confidence score of all participants was higher immediately after the course in both case and image questions (*p* < 0.001 for both) (Figure 3a). When the increase in confidence was analyzed, stratified by experience for those returning both questionnaires, it showed a higher increase for beginners across both parts of the assessment both in post- and follow-up tests. For the other groups, change in confidence levels remained statistically non-significant in follow-up test compared to the pre-course level, which might also be due to the low number of follow-up test takers in the intermediate and experts’ groups (Figure 3b,c).

Based on our analysis, we divided the participants into the groups: low-confidence/low-score, low-confidence/high-score, high-confidence/low-score and high-confidence/high-score. A significant proportion of the participants belonged to the high-confidence/low-score group in the pre-test in all experience groups, but the proportion was lower after the course for all groups (Table 3).

## 3. Discussion

We studied the effect of a case-based course on colposcopy in three European countries and discovered that a one-day course using patient cases, teacher-supported active problem solving and guided image-recognition could substantially improve colposcopy knowledge and diagnostic skills.

These results were consistent with the results of a previous study that suggested improvement in the ability to manage colposcopy cases. They detected high heterogeneity in the results across countries. The study did not examine the level of experience of the participants [5], which we found in this study to affect the test results. We also separately assessed whether the course had contributed to the solution of case-based scenarios and image-recognition and whether the improvement was sustainable over time.

The effect of the course on the post-course scores was more pronounced in the section including patient cases. Significant improvement in image recognition was observed, but to a lesser extent. There was no improvement in the recognition of high-grade versus low-grade lesions and normal versus abnormal lesions after the course. This suggests that image analysis and colposcopic diagnostics might require more training to achieve similar outcomes. Learning to distinguish between high-grade and low-grade lesions is important to emphasize when further developing the course.

The improvement in overall performance occurred in all groups, but most notably among the beginners. The scores for beginners, two months after the course were as high as the scores of the experts before the course. A study in radiological diagnostics by Lesgold et al. showed that beginners had not yet developed the fine-tuned visual perception to perform successful feature discrimination in visual analysis [18]. This might be one reason why a systematic image analysis, such as that used in our course, improved the performance of the beginners.

The aim of the training was to improve both performance and confidence of the participants regarding the colposcopies. However, we saw that a substantial proportion of participants had high confidence but poor performance—not a favorable outcome, as this combination might lead to reduced patient safety [19]. Such a combination was most prominent among those who had intermediate experience, but this could be identified in all groups. The overall number of participants with high confidence and poor scores reduced after the course. The appearance of overconfidence was consistent with the result of other studies that also show a larger risk of on diagnostic overconfidence occurring among residents compared to medical students and consultants [19].

Previous studies suggest that the mode of clinical reasoning among beginners and intermediate physicians is largely hypotheticodeductive (seeking evidence to evaluate the original hypothesis). This method is more prone to premature closure (the case is closed when a suitable solution is found ignoring other possibilities) or confirmation bias (tendency to look for information that confirms the original idea), leading to an illusion of competence. Experts tend to solve difficult problems mainly using pattern-recognition, which requires a wide repertoire of experienced cases. Experts are also more prone to recognize if the initial impression was wrong. However, overconfidence can also occur in pattern-recognition, either related to insufficient clinical knowledge or to shortcomings of heuristic problem solving (i.e., subconscious conclusions) [20,21,22,23].

This is the first study to describe the impact of structured case-based and interactive colposcopy courses on the ability to manage colposcopy cases and the diagnostic accuracy of image recognition. A large number of participants with different levels of experience from three countries participated the study. The study assessed the development of the participants, whether the improvement was sustainable in the longer term (two months) and how the development of the level of knowledge correlated with confidence in one’s own competence.

The study has some limitations and caution is advised in interpreting the results. Only one-third of participants returned the follow-up test after two months, and the related analyses lack the statistical power to demonstrate significant differences; also, the low participation increases the risk of selection bias, which may outweigh the improvement in outcomes. Therefore, the conclusions that can be drawn on the results from the follow-up test on long-term learning are limited. Another limitation is that we did not collect information on the medical profession of the participants (specialists, residents, medical students or nurse practitioners). However, the aim of the study was to evaluate the effect of the course. Since we collected the baseline knowledge of the participants in the pre-test, we were able to evaluate this regardless of the profession. Thus, we do not see this as a major problem.

Furthermore, the two-month follow-up test was a web-based format with no time limit, which allowed more time to complete the test, which makes the results not entirely comparable with the pre- and post-tests. The images used in the assessment were not part of the course training-material. However, the case scenarios were used during the course, and therefore some scores might reflect the ability to remember correct answers rather than independent problem-solving. The cases were presented in the same order, which might again have the same effect. We also did not know how many colposcopies the participants performed between the course and follow-up test. However, two months is a very short time to achieve major experience in colposcopy so the effect of clinical practice in the meantime, on the results of the follow-up test, is considered small. Another limitation was that we only had one still image of the cervix per case, which does not correspond to the dynamic nature of colposcopy. The content of the course was basic knowledge and skills, which would increase learning among beginners more than among experienced participants.

Although the dropout rate of our study in the two-month follow-up test was high, the study showed that a well-structured course that supported learners’ active learning by using authentic patient cases and digital images as triggers for learning proved to be a useful model for course design for colposcopes at all competence levels. It is especially suitable for those with limited colposcopy experience. In this case, teacher-supported clinical reasoning tasks were important for learning. These colposcopists still require in-service training and need to learn to assess their true skills because some of the participants showed high confidence combined with poor results. Educators and supervising physicians should be aware of this phenomenon as the risk of medical errors may increase. Hands-on training is also needed to complement the theoretical training of colposcopists. An interesting subject for further research could be the evaluation of the clinical impact of the course.

## 4. Materials and Methods

We performed a prospective multicenter interventional pilot study on three one-day colposcopy courses in 2015 under the auspices of the EFC in Finland, Norway and the United Kingdom. All courses had an identical course design, the same teachers, and the courses were open to all health care professionals [1]. The courses included a six-hour teaching session covering the etiology and epidemiology, diagnosis and management of cervical pre-cancer, including analysis of several clinical cases and colposcopic images together with the participants (for a detailed course curriculum, see Appendix A). To increase active learning, the participants were encouraged to discuss, ask questions and participate in the learning activities [24]. The aim was to teach the participants structured inspection and pattern recognition of a colposcopic image, including determining the type of uterine cervical transformation zone (TZ) and occurrence of abnormality, selecting the optimal biopsy location, and the correct diagnosis of the abnormality. Participants were asked to speak aloud through the problem-solving process while defining step-by-step the Reid Colposcopy Index (RCI) criteria [25,26,27] based on three different morphological features. Teachers asked supporting questions until the participant was able to proceed independently or the image was analyzed together. This structured method of analyzing the image is recommended for routine use in colposcopies.

All attendants were invited to participate in the study. A questionnaire was distributed before and after the completion of the course, and two months later with a third follow-up web-based questionnaire (made with Webropol web-based questionnaire tool [28]). The tests were divided into two parts and the questions were identical at all times. The follow-up time was limited to two months, a time-frame within which we expected both reaching the participants as well as assessing the efficacy of the course would still be possible [29]. The tests were provided in the official language of the participating countries.

The first part of each test included ten colposcopic cases, presenting women with cervical abnormalities and multiple-choice questions on the appropriate management strategies (Appendix A). The same case-questions were used in the study done by Zahid and colleagues [9]. The maximum score was ten (one point per correct answer); in the case of a mix of right and wrong answers, the answer was scored a half-point. The correct answers (one or more) were decided upon by an expert panel consisting of five senior colposcopists from three countries.

The second part consisted of ten colposcopic digital images (10–16× magnification, whole uterine cervix visible, shown for 30 s on a screen, and unlimited time in a web-based follow-up test) of normal and abnormal cervices. The participants were asked to determine the TZ type according to the IFCPC (the International Federation of Cervical Pathology and Colposcopy) nomenclature [30] and to choose a colposcopic diagnosis from the responses to multiple-choice questions, each of which may receive a maximum of 10 points. (Appendix A). Participants were unaware of Pap-smear and HPV results. The diagnosis-identification was further divided into subcategories: correct identification of high/low-grade lesion (high-grade lesions including CIN2, CIN3, microinvasive squamous carcinoma, macroscopic squamous cell carcinoma (SCC) and adenocarcinoma), and normal/abnormal lesion, with a maximum of ten points each. Two of the images presented invasive cancer and correct identification of invasion resulted in one point in an additional subcategory with up to two points. The maximum number of points in the whole image recognition part was, therefore, ten points in four categories and two points in one (42 points). There were no negative grades for incorrect replies. In each image, the correct answer (only one) was defined by the result of histopathological analysis of the biopsy.

In each question, participants determined how confident they were in their answer on a 100 mm visual analogue scale (VAS). The extremes were “very confident” (100 mm) and “not at all confident” (0 mm). We further assessed the confidence levels confidence separately for the two parts, as reported by the participants using the mean value of VAS scores across all answered questions for each participant. Unanswered questions were removed from the analysis. We analyzed how the confidence level correlated with the rates of correct responses using the Pearson’s correlation test. To assess the degree of confidence, the participants were divided into four groups with combinations of low or high score and low or high confidence. As cut-off-point, we used 15 points (half of the maximum of 30 scores in main categories: correct treatment or follow up, correct TZ-type and correct diagnosis, added together) and 100 mm (half of the maximum mean VAS score in case and image questions added together).

We performed further subgroup analyses to stratify the participants by experience level based on the annual number of colposcopies performed (defined as beginners (<50 colposcopies), intermediate (50–200 colposcopies) and expert colposcopists (>200 colposcopies)).

The data were analyzed using the SPSS software (IBM Corp. Released 2013. IBM SPSS Statistics for Windows, Version 22.0. Armonk, NY, USA). All statistical tests used were two-tailed with the level of statistical significance at *p* < 0.05.

This study concerned the training of medical professionals and was conducted without patient involvement. We consulted the Ethics Committee for Gynecology and Obstetrics, Pediatrics and Psychiatry at The Hospital District of Helsinki and Uusimaa, Finland, concerning the need for an ethical approval for the study. The reply of the representative of the committee was that since the study does not include any patients, no approval is needed. The committee also stated that the study did not require a written consent of the participants. All participants gave oral consent to participate in the study.

## 5. Conclusions

A structured case-based course based on active learning methods improves skills and confidence, especially for inexperienced colposcopists. However, teachers should be aware of the risk of overconfidence, which seems to be prevalent especially in some of the intermediate colposcopists. The impact of the course was highest on case management but still present on colposcopic diagnosis. To complement the theoretical training, a practical internship with high quality feedback is recommended. Due to the small number of participants in the follow-up study, the conclusions on long-term learning are limited. An interesting topic for further research would be to assess the impact of the course on clinical competence.

## Figures and Tables

**Figure 1 cancers-12-03201-f001:**
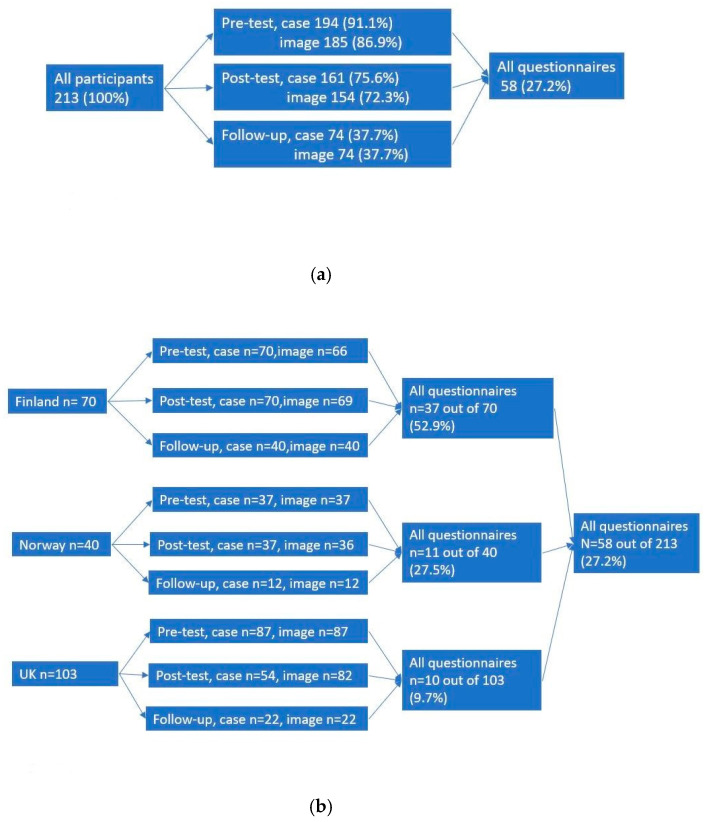
Returned questionnaires in pre-, post- and follow up test (**a**) and according to countries (**b**).

**Figure 2 cancers-12-03201-f002:**
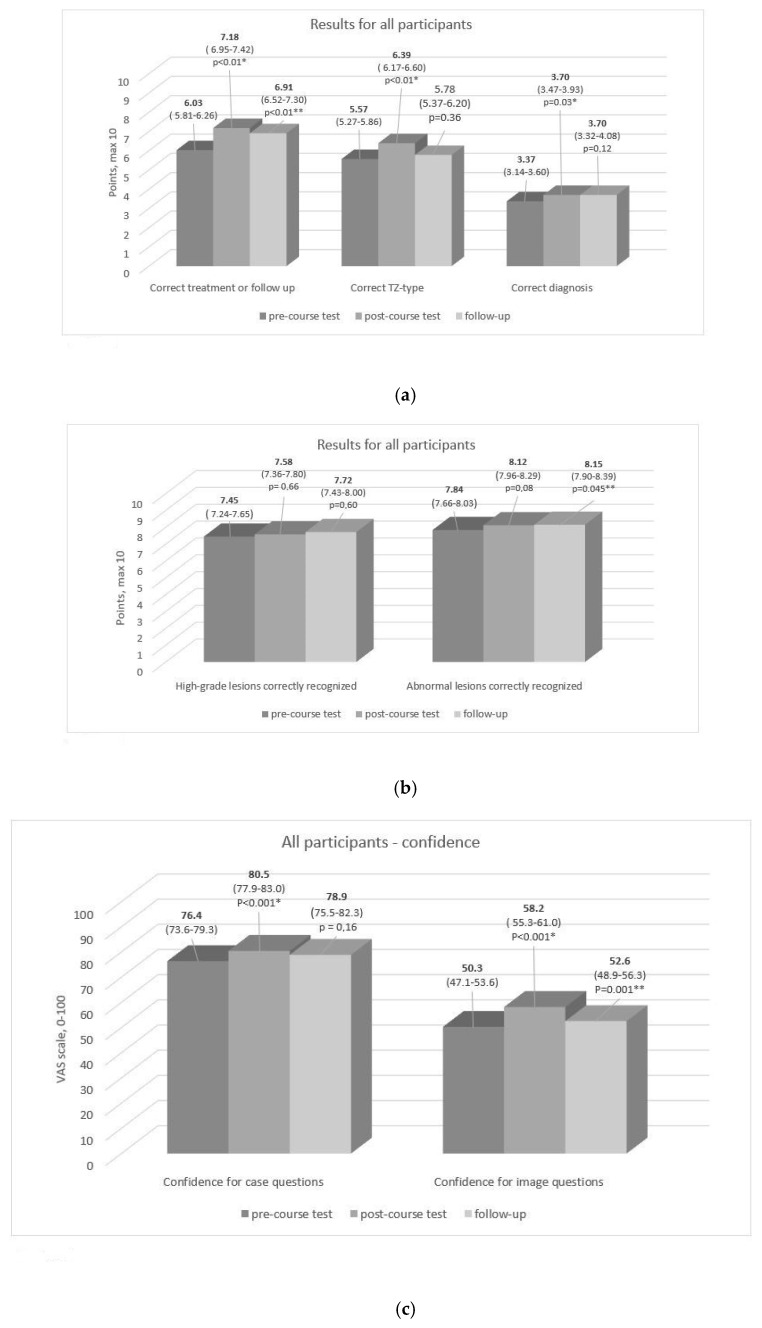
Mean values for points (out of ten) in categories correct treatment or follow-up, correct TZ and correct diagnosis (**a**), categories high-grade lesion recognized and abnormal lesion recognized (**b**), and values for confidence (0 = not at all confident, 100 = very confident) (**c**) for all participants. *p*-Value is presented if the change in points or confidence pre- to post-course test (marked with *), and pre-course to follow up (marked with**) is statistically significant. Numbers in parentheses indicate 95% confidence interval.

**Figure 3 cancers-12-03201-f003:**
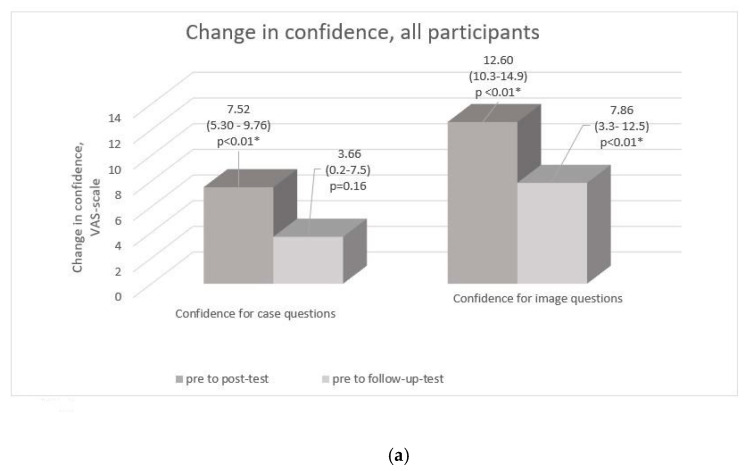
Results stratified by experience. Change in mean values for confidence (0 = not at all confident, 100 = very confident) in all questions (**a**) and separately for case questions (**b**) and image questions (**c**) according to experience. *p*-Value is presented with * if the change is statistically significant.

**Table 1 cancers-12-03201-t001:** Demographic information about the participants.

Basic Information		Finland	Norway	UK	All
Mean age (years)		43.4	39.0	49.0	45.0
Sex					
Female (*n*)		63 (90.0%)	30 (75.0%)	58 (56.3%)	151 (70.9%)
Male (*n*)		6 (8.6%)	7 (17.5%)	29 (28.2%)	42 (19.7%)
Sex not known (*n*)		1 (1.4%)	3 (7.5%)	16 (15.5%)	20 (9.4%)
All		70 (100%)	40 (100%)	103 (100%)	213 (100%)
Mean experience of colposcopies (years)		10.41	5.12	12.33	10.33
Experience of colposcopies	Not known	1 (1.4%)	3 (7.5%)	17 (16.5%)	21 (9.9%)
	Beginner/low caseload, <50 colposcopies/year	42 (60.0%)	25 (62.5%)	7 (6.8%)	74 (34.7%)
	Intermediate, 50–200 colposcopies/year	20 (28.6%)	9 (22.5%)	51 (49.5%)	80 (37.6%)
	Expert/high caseload >200 colposcopies/year	7 (10.0%)	3 (7.5%)	28 (27.2%)	38 (17.8%)
	Total	70 (100%)	40 (100%)	103(100%)	213 (100%)

**Table 2 cancers-12-03201-t002:** Mean values and 95% confidence intervals (CI) for points in case questions and image.

Time of Test		Correct Treatment or Follow Up/10 Points		Correct TZ Type/10 Points		Correct Diagnosis/10 Points		High/Low Grade Lesions Correctly Recognized/10 Points		Abnormal/Normal Lesions Correctly Recognized/10 Points	
		95% CI		95% CI		95% CI		95% CI		95% CI	
Beginners	pre-course test	5.68	5.28–6.09	5.67	5.08–6.23	3.35	2.99–3.71	7.20	6.79–7.62	7.83	7.44–8.21
	post-course test	7.28 ^1^	6.92–7.65	6.64 ^2^	6.30–6.99	3.90 ^4^	3.56–4.23	7.54	7.25–7.82	8.24	7.99–8.49
	follow-up	6.83 ^1^	6.19–7.47	5.91	5.37–6.46	3.69	3.09–4.29	7.63	7.22–8.04	8.11	7.74–8.50
Intermediate	pre-course test	6.10	5.76–6.43	5.34	4.87–5.82	3.36	2.98–3.74	7.57	7.30–7.84	7.84	7.58–8.10
	post-course test	7.03 ^1^	6.68–7.38	6.29 ^3^	5.92–6.67	3.62	3.21–4.03	7.54	7.09–7.98	7.98	7.69–8.28
	follow-up	6.83	6.22–7.44	5.58	4.68–6.48	3.62	3.07–4.16	7.73	7.38–8.08	8.31 ^5^	7.99–8.63
Experts	pre-course test	6.59	6.12–7.07	5.79	5.30–6.27	3.72	3.09–4.35	7.82	7.49–8.14	7.97	7.64–8.30
	post-course test	7.26 ^2^	6.64–7.91	6.08	5.54–6.62	3.67	2.93–4.40	7.92	7.35–8.49	8.25	7.83–8.67
	follow-up	7.18	6.22–8.15	6.00	5.33–6.67	4.18	2.84–5.52	8.18	6.91–9.45	8.09	7.22–8.96

Significant increase in point as compared to pre-test, *p*-values for Wilcoxon test: ^1^
*p* < 0.001, ^2^
*p* = 0.02, ^3^
*p* = 0.004, ^4^
*p* = 0.04, ^5^
*p* = 0.006.

**Table 3 cancers-12-03201-t003:** Confidence in answers in all three tests, stratified according to clinical experience.

Participant Group		Confidence					
		Low = VAS < 100, High = VAS ≥ 100					
		Pre-Test		Post Test		Follow-Up Test	
	All participants	Low (*n*)	High (*n*)	Low (*n*)	High (*n*)	Low (*n*)	High (*n*)
	High (*n*)	18 (10.9%)	81 (49.1%)	16 (11.5%)	103 (74.1%)	4 (5.4%)	45 (60.8%)
	Low (*n*)	22 (13.3%)	44 (26.7%)	2 (1.4%)	18 (12.9%)	4 (5.4%)	21 (28.4%)
			Total 165 (100%)		Total 139 (100%)		Total 74 (100%)
	Beginners	Low (*n*)	High (*n*)	Low (*n*)	High (*n*)	Low (*n*)	High (*n*)
	High (*n*)	15 (23.8%)	22 (34.6%)	15 (23.8%)	42 (66.7%)	3 (8.6%)	20 (57.1%)
Scores	Low (*n*)	16 (25.4%)	10 (15.9%)	1 (1.6%)	5 (7.9%)	4 (11.4%)	8 (22.9%)
			Total 63 (100%)		Total 63 (100%)		Total 35 (100%)
High ≥ 15 points	Intermediate	Low (*n*)	High (*n*)	Low (*n*)	High (*n*)	Low (*n*)	High (*n*)
Low <15 points	High (*n*)	1 (1.5%)	37 (55.2%)	0	36 (81.8%)	0	17 (65.4%)
	Low (*n*)	5 (7.5%)	24 (35.8%)	0	8 (18.2%)	0	9 (34.6%)
			Total 67 (100%)		Total 44 (100%)		Total 26 (100%)
	Experts	Low (*n*)	High (*n*)	Low (*n*)	High (*n*)	Low (*n*)	High (*n*)
	High (*n*)	1 (3.1%)	22 (68.8%)	0	20 (83.3%)	0	8 (72.7%)
	Low (*n*)	1 (3.1%)	8 (25.0%)	1 (4.2%)	3 (12.5%)	0	3 (27.3%)
			Total 32 (100%)		Total 24 (100%)		Total 11 (100%)

Participants were divided in groups according to high (VAS ≥ 100) or low (VAS < 100) confidence and high scores (points in categories treatment and follow-up, TZ and diagnosis added together ≥ 15) or low scores (points in categories treatment and follow-up, TZ and diagnosis added together <15). The division was done for all participants and stratified according to experience.

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
