# Peer review of "Fostering Prevention of Cervical Cancer by a Correct Diagnosis of Precursors: A Structured Case-Based Colposcopy Course in Finland, Norway and UK"

_cancers, 2020, doi:10.3390/cancers12113201_

Round 1

Reviewer 1 Report

I have read this work with great interest and thank you for having had the opportunity to revise it.

it would be interesting to know if it was more difficult for colposcopists to recognize a lesion of the cervix or vagina, which was the detecton rate of the lesions, there is a region of the lower genital tract whose lesions were more difficult to explore.Is it possible to identify from your questionnaire if images captured in a particular phase of the menstrual cycle or in a particular age of the woman or period of pregnancy were correlated to a worse detection rate?Does your study show a solution to improve the diagnostic capacity especially for beginner colposcopists?

Author Response

I have read this work with great interest and thank you for having had the opportunity to revise it.

Answer: Thank you for your interest in our work and your valuable comments

it would be interesting to know if it was more difficult for colposcopists to recognize a lesion of the cervix or vagina, which was the detection rate of the lesions, there is a region of the lower genital tract whose lesions were more difficult to explore.

Answer: Thank you for your comment. We agree that this is a very interesting issue. However, this study focused on cervical lesions and we have not therefore been able to study the diagnostic accuracy on vaginal lesions. This is an interesting topic for a future study.

Is it possible to identify from your questionnaire if images captured in a particular phase of the menstrual cycle or in a particular age of the woman or period of pregnancy were correlated to a worse detection rate?

Answer: Thank you for your comment. None of the patients was pregnant nor postmenopausal. All patients were between 18 and 50 years of age. State of menstrual cycle or use of contraception were not recorded. Although this might affect the accuracy of colposcopy in general, we do not consider the lack of this information to affect the interpretation of our results here.  

Does your study show a solution to improve the diagnostic capacity especially for beginner colposcopists?

Answer: Thank you for your question. Based on our results a colposcopy course with an activating didactic method and case-based curriculum can be of help in increasing the basic diagnostic capacity of beginner colposcopists. However, we believe that a course has to be complemented with continuous feedback and hands-on teaching and training in clinic.

We address this matter in the discussion chapter page 8, lines 213-221:

“…, the study showed that a well-structured course that supported learners’ active learning by using authentic patient cases and digital images as triggers for learning proved to be a useful model for course design for colposcopes at all competence levels. It is especially suitable for those with limited colposcopy experience. In this case, teacher-supported clinical reasoning tasks were important for learning. These colposcopists still require in-service training and need to learn to assess their true skills because some of the participants showed great confidence combined with poor results. Educators and supervising physicians should be aware of this phenomenon as the risk medical errors may increase. Hands-on training is also needed to complement the theoretical training of colposcopists….”

Reviewer 2 Report

This manuscript presents results of a one-day colposcopy training course. The research involves 213 colposcopists from three countries with a variety of experience and training. The survey design including pre, post and follow-up questions was well constructed.

Research involving colposcopy accuracy is very challenging because of the inherent variability of reviews. Still, training is required and should be well-documented. Unfortunately, each colposcopist reviewed only 10 cases. How was this number decided upon? I'm concerned that it's not sufficient, especially given that the colposcopists evaluated the same cases each time and likely remembered their previous response. 

Follow-up questionnaires were answered by a small proportion of colposcopists and interpretation of those results should be limited.

The analysis is and paper overall are too long and it is recommended that the authors rewrite the paper and focus on the most pertinent and critical findings.   

Author Response

Reviewer 2.

This manuscript presents results of a one-day colposcopy training course. The research involves 213 colposcopists from three countries with a variety of experience and training. The survey design including pre, post and follow-up questions was well constructed.

Research involving colposcopy accuracy is very challenging because of the inherent variability of reviews. Still, training is required and should be well-documented. Unfortunately, each colposcopist reviewed only 10 cases. How was this number decided upon? I'm concerned that it's not sufficient, especially given that the colposcopists evaluated the same cases each time and likely remembered their previous response. 

Answer: Thank you for this comment. Our study was incorporated in to a tight course scheduled of six hours. Altogether 10 different images and 10 different patient cases were considered to be an absolute maximum regarding the schedule.  During the course the participants saw few hundred different colposcopy images and remembering those 10 test-images was not considered to be a disadvantage. Also changing the test images would have reduced the chances of testing equally, without changing the degree of difficulty.

We have also in the discussion chapter addressed the risk of the results partly reflecting the ability to remember rather than independent problem solving. Page 8 lines 204-206 reads:

“However, the case scenarios were used during the course, and therefore some scores might reflect the ability to remember correct answers rather than independent problem-solving. The cases were presented in the same order, which might again have the same effect.”

Follow-up questionnaires were answered by a small proportion of colposcopists and interpretation of those results should be limited.

Answer: Thank you for this important comment. We agree that the low participation in the follow-up test is a limitation and that the conclusions that can be drawn based on the results from that test is roughly limited.  We also believe that there is a risk of a selection bias. We have mentioned this limitation in the discussion (page 8 line 192- ) and conclusion chapters (page 10 lines 296-297 ) and the abstract (page 1 lines 46-47)

We have now further amended the discussion section to further emphasize that great caution is needed when interpreting the results from the follow-up test and drawing any conclusions based on them. Page 9 lines 192-196 now reads:

“The study has some limitations and caution is advised in interpreting the results. Only one-third of participants returned the follow-up test after two months, and the related analyses lack the statistical power to demonstrate significant differences, also the low participation increase the risk of selection bias, that may outweigh the improvement in outcomes. Therefore, the conclusions that can be drawn on the results from the follow-up test on long-term learning is limited.”

The analysis is and paper overall are too long and it is recommended that the authors rewrite the paper and focus on the most pertinent and critical findings.  

Answer: Thank you for the advice. We have shortened the manuscript and we have also edited the tables 2 and removed figure 4. In case further shortening of the manuscript is warranted, we are naturally happy to comply.

Reviewer 3 Report

High-quality colposcopy is essential in cervical cancer prevention. The aim of the European Federation of Colposcopy is to improve the education and training in colposcopy, e.g. by organizing colposcopy courses. Sabrina et al. performed a prospective interventional pilot study, evaluating the effect of a six-hour colposcopy course. A total of 213 colposcopists took part in the study (70 in Finland, 40 in Norway, 103 in the UK). The mean score in the case questions increased. The image recognition scores of some of the evaluated elements were higher immediately after the course compared to the scores obtained before the course. In the subcategories (high- vs low-grade lesion and normal vs. abnormal lesion recognition), there was no improvement after the course, but the detection of abnormal lesion was better in the follow-up test compared to the pre-course test. A structured case-based course improves skills and confidence especially for inexperienced colposcopists; however, trainers should be aware of the risk of overconfidence. To complement theoretical training, further hands-on training including high-quality feedback is recommended. Conclusions drawn from long-term learning are limited due to the low participation in the follow up test.

The claims are properly placed in the context of the previous literature. The experimental data support the claims. The manuscript is written clearly enough that most of it is understandable to non-specialists. The authors have provided adequate proof for their claims, without overselling them. The authors have treated the previous literature fairly. The paper offers enough details of methodology so that the experiments could be reproduced.

Comments

Colposcopy is subjective and has limited sensitivity for high grade lesions (CIN2+). The National Health Service Cervical Screening Programme (NHSCSP) Guidelines for Colposcopy and Programme Management, which guides British practice, ask for evidence of a colposcopic accuracy of 65% (Desai 2010). Zuchna et al reported 66.2% sensitivity of CIN2+ when up to three guided cervical biopsies were taken regarded as a diagnostic test with the cone specimen as reference standard (Zuchna 2010). Using digitized cervical images from 919 women referred for equivocal or minor cytologic abnormalities into the ASCUS-LSIL Triage Study, Massad et al reported 39% sensitivity for CIN2+ (Massad 2009). Regardless of skill, performing more biopsies increases the sensitivity of colposcopy (Pretorius 2011). Dynamic Spectral Imaging System (DySIS) colposcopy seemed inferior to conventional colposcopy in detecting high-grade lesions and cannot replace conventional colposcopy with random biopsies (Roensbo 2015). The observed high sensitivity of the punch biopsy derived from all studies is probably the result of verification bias (Underwood 2012). In a study from Norway, women with negative or low-grade cervical biopsies (normal/CIN1) were followed up after six months in order to decide on further follow-up or recall for screening at three-year intervals. Of 520 women with negative or low-grade biopsy, 124 women (23.8%) had CIN2+ in follow-up biopsy, including 7 cases of invasive cervical cancer (Sorbye 2011). Hence, all women with negative colposcopy and biopsies after abnormal cytology and/or HPV-testing have to be followed. 

Page 1, line 2-4, title "Fostering early diagnosis of cervical cancers: A structured case-based colposcopy course in Finland, Norway and UK"

The concept of screening is not to detect prevalent cervical cancer, but identifying and treating the precancerous lesions before development of invasive cervical carcinoma.

Page 1, line 24-26, "Colposcopy plays a major role in this management, not the least in developing countries were cervical cancer is more prevalent and access to vaccines still will be difficult for a large part of the population."

I disagree. In most developing countries colposcopy is not an option. HPV-screening and cervical cytology is usually not available. The visual inspection techniques include VIA (visual inspection with acetic acid) and VILI (visual inspection using Lugol's iodine). These approaches are an attractive alternative to cytology-based screening in LMIC. Often screen-and-treat strategies involve treatment with cryotherapy without colposcopy and biopsy.

Page 1, line 39-40, "Outcome measures where mean scores in correct case-management, diagnosis, transformation-zone recognition and confidence in answers"

Is it possible to say something about sensitivity for CIN2+ before and after the course?

Page 2, line 48-55, delete "A single paragraph of about 200 words maximum. For research articles, abstracts should give a pertinent overview of the work. We strongly encourage authors to use the following style of structured abstracts, but without headings: (1) Background: Place the question addressed in a broad context and highlight the purpose of the study; (2) Methods: Describe briefly the main methods or treatments applied; (3) Results: Summarize the article's main findings; and (4) Conclusions: Indicate the main conclusions or interpretations. The abstract should be an objective representation of the article, it must not contain results which are not presented and substantiated in the main text and should not exaggerate the main conclusions."

Page 2, Introduction, It is possible to say something about the limited sensitivity of colposcopy regarding CIN2+ (or CIN3+).

Page 2, Results, Is it possible to say something about sensitivity for CIN2+ before and after the course?

Page 3, Table 1, The percentages do not match 100% (eg 91.3% + 8.7% + 1.4% = 101.4%)

Page 4, Table 2, This table is too complex with too much information. Drop the 95% CI and p-values in the table. Results that are statistical significant can be marked with an asterix (*) and a footnote (p<0.01). Results from "Intermediate" and "Experts" could be combined in one group. You can also drop the column "Invasion recognized".

Page 6, Figure 3, Results from "Intermediate" and "Experts" could be combined in one group.

Page 7, Figure 4, Results from "Intermediate" and "Experts" could be combined in one group.

Page 7, Table 3, Results from "Intermediate" and "Experts" could be combined in one group.

Page 8, Discussion, Is it possible to say something about sensitivity for CIN2+ before and after the course?

References

Desai M, Hadden P, Kitchener H, Martin-Hirsch P, Prendiville W, et al. (2010) Colposcopy and Programme Management. Guidelines for the NHS Cervical Screening Programme. NHSCSP Publication No 20 May.

Zuchna C, Hager M, Tringler B, Georgoulopoulos A, Ciresa-Koenig A, et al. (2010) Diagnostic accuracy of guided cervical biopsies: a prospective multicenter study comparing the histopathology of simultaneous biopsy and cone specimen. Am J Obstet Gynecol 203: 321–326.

Massad LS, Jeronimo J, Katki HA, Schiffman M (2009) The accuracy of colposcopic grading for detection of high-grade cervical intraepithelial neoplasia. J Low Genit Tract Dis 13: 137–144.

Pretorius RG, Belinson JL, Burchette RJ, Hu S, Zhang X, Qiao YL. Regardless of skill, performing more biopsies increases the sensitivity of colposcopy. J Low Genit Tract Dis. 2011 Jul;15(3):180-8. doi: 10.1097/LGT.0b013e3181fb4547. PMID: 21436729.

Roensbo MT, Hammer A, Blaakaer J. Can Dynamic Spectral Imaging System colposcopy replace conventional colposcopy in the detection of high-grade cervical lesions? Acta Obstet Gynecol Scand. 2015 Jul;94(7):781-785.

Underwood M, Arbyn M, Parry-Smith W, De Bellis-Ayres S, Todd R, Redman CW, Moss EL. Accuracy of colposcopy-directed punch biopsies: a systematic review and meta-analysis. BJOG. 2012 Oct;119(11):1293-301.

Sorbye SW, Arbyn M, Fismen S, Gutteberg TJ, Mortensen ES. HPV E6/E7 mRNA testing is more specific than cytology in post-colposcopy follow-up of women with negative cervical biopsy. PLoS One. 2011;6(10):e26022.

Author Response

Reviewer 3.

High-quality colposcopy is essential in cervical cancer prevention. The aim of the European Federation of Colposcopy is to improve the education and training in colposcopy, e.g. by organizing colposcopy courses. Sabrina et al. performed a prospective interventional pilot study, evaluating the effect of a six-hour colposcopy course. A total of 213 colposcopists took part in the study (70 in Finland, 40 in Norway, 103 in the UK). The mean score in the case questions increased. The image recognition scores of some of the evaluated elements were higher immediately after the course compared to the scores obtained before the course. In the subcategories (high- vs low-grade lesion and normal vs. abnormal lesion recognition), there was no improvement after the course, but the detection of abnormal lesion was better in the follow-up test compared to the pre-course test. A structured case-based course improves skills and confidence especially for inexperienced colposcopists; however, trainers should be aware of the risk of overconfidence. To complement theoretical training, further hands-on training including high-quality feedback is recommended. Conclusions drawn from long-term learning are limited due to the low participation in the follow up test.

The claims are properly placed in the context of the previous literature. The experimental data support the claims. The manuscript is written clearly enough that most of it is understandable to non-specialists. The authors have provided adequate proof for their claims, without overselling them. The authors have treated the previous literature fairly. The paper offers enough details of methodology so that the experiments could be reproduced.

Comments

Colposcopy is subjective and has limited sensitivity for high grade lesions (CIN2+). The National Health Service Cervical Screening Programme (NHSCSP) Guidelines for Colposcopy and Programme Management, which guides British practice, ask for evidence of a colposcopic accuracy of 65% (Desai 2010). Zuchna et al reported 66.2% sensitivity of CIN2+ when up to three guided cervical biopsies were taken regarded as a diagnostic test with the cone specimen as reference standard (Zuchna 2010). Using digitized cervical images from 919 women referred for equivocal or minor cytologic abnormalities into the ASCUS-LSIL Triage Study, Massad et al reported 39% sensitivity for CIN2+ (Massad 2009). Regardless of skill, performing more biopsies increases the sensitivity of colposcopy (Pretorius 2011). Dynamic Spectral Imaging System (DySIS) colposcopy seemed inferior to conventional colposcopy in detecting high-grade lesions and cannot replace conventional colposcopy with random biopsies (Roensbo 2015). The observed high sensitivity of the punch biopsy derived from all studies is probably the result of verification bias (Underwood 2012). In a study from Norway, women with negative or low-grade cervical biopsies (normal/CIN1) were followed up after six months in order to decide on further follow-up or recall for screening at three-year intervals. Of 520 women with negative or low-grade biopsy, 124 women (23.8%) had CIN2+ in follow-up biopsy, including 7 cases of invasive cervical cancer (Sorbye 2011). Hence, all women with negative colposcopy and biopsies after abnormal cytology and/or HPV-testing have to be followed. 

Page 1, line 2-4, title "Fostering early diagnosis of cervical cancers: A structured case-based colposcopy course in Finland, Norway and UK"

The concept of screening is not to detect prevalent cervical cancer, but identifying and treating the precancerous lesions before development of invasive cervical carcinoma.

Answer: Thank you for your comment. We completely agree with the referee and have changed the title of the article to “Fostering prevention of cervical cancer by a timely diagnosis of precursors: A structured case-based colposcopy course in Finland, Norway and UK”. "

Page 1, line 24-26, "Colposcopy plays a major role in this management, not the least in developing countries were cervical cancer is more prevalent and access to vaccines still will be difficult for a large part of the population."

I disagree. In most developing countries colposcopy is not an option. HPV-screening and cervical cytology is usually not available. The visual inspection techniques include VIA (visual inspection with acetic acid) and VILI (visual inspection using Lugol's iodine). These approaches are an attractive alternative to cytology-based screening in LMIC. Often screen-and-treat strategies involve treatment with cryotherapy without colposcopy and biopsy.

Answer: Thank you for this comment. We agree and have removed those sentences on lines 24-26.

Page 1, line 39-40, "Outcome measures where mean scores in correct case-management, diagnosis, transformation-zone recognition and confidence in answers"

Is it possible to say something about sensitivity for CIN2+ before and after the course?

Answer:  Thank you for this relevant question. We have studied the effect on the course on recognition of high-grade lesions (CIN2+). This is now also mentioned in the abstract. Page 1 lines 38-39 now read:

“Outcome measures where mean scores in correct case-management, diagnosis (including high-grade lesion recognition), transformation-zone recognition and confidence in answers.”

Page 2, line 48-55, delete "A single paragraph of about 200 words maximum. For research articles, abstracts should give a pertinent overview of the work. We strongly encourage authors to use the following style of structured abstracts, but without headings: (1) Background: Place the question addressed in a broad context and highlight the purpose of the study; (2) Methods: Describe briefly the main methods or treatments applied; (3) Results: Summarize the article's main findings; and (4) Conclusions: Indicate the main conclusions or interpretations. The abstract should be an objective representation of the article, it must not contain results which are not presented and substantiated in the main text and should not exaggerate the main conclusions."

Answer: Thank you very much for pointing this out. The text from the template provided by the journal accidentally remained in the submitted version and has now been removed.

Page 2, Introduction, It is possible to say something about the limited sensitivity of colposcopy regarding CIN2+ (or CIN3+).

Answer: Thank you for this relevant question. The sensitivity of colposcopy in finding CIN 2+ lesion has been a hot topic for a longer time and a quite a few articles have been published dealing with this showing remarkably different outcomes, as pointed out by the referee as well.  This study also include evaluation of the effect on the course on recognition of high-grade lesions (CIN2+). We have added a section on this also in the introduction chapt

 Page 2 lines 77-82 now read:

“The aim of our study was to evaluate whether this standardized basic case-based colposcopy course, improved the ability to manage colposcopy patient cases, diagnostic accuracy and confidence. Many studies have shown that colposcopy has a limited sensitivity for high-grade lesions.In this study we also evaluated the effect of the course on recognition of high-grade (CIN2+) vs low-grade lesion and normal vs. abnormal lesions.”

We also thank you for the relevant references on this issue

Page 2, Results, Is it possible to say something about sensitivity for CIN2+ before and after the course?

Answer: Please see also the answers to the previous questions on the same issue. The results show that the sensitivity to diagnose CIN 2+ lesion is not significantly improved after the course.  This issue is addressed in the results section page 2, lines 91-93, figure 2b and table 2.

Page 3, Table 1, The percentages do not match 100% (eg 91.3% + 8.7% + 1.4% = 101.4%)

Answer: Thank you for this comment. The percentages have been recalculated and the table has been amended accordingly.

Page 4, Table 2, This table is too complex with too much information. Drop the 95% CI and p-values in the table. Results that are statistical significant can be marked with an asterix (*) and a footnote (p<0.01). Results from “Intermediate” and “Experts” could be combined in one group. You can also drop the column “Invasion recognized”.

Answer: The table 2 has now been reconstructed according to the comment. However, we would not like to merge intermediate and expert groups because essential data would be lost.  In case this is considered crucial by the reviewer or the editorial board, we are naturally happy to comply.

Page 6, Figure 3, Results from "Intermediate" and "Experts" could be combined in one group.

Page 7, Figure 4, Results from "Intermediate" and "Experts" could be combined in one group.

Page 7, Table 3, Results from "Intermediate" and "Experts" could be combined in one group.

Answer:  We feel that these groups should not be merged. Essential information would be lost. There is a clear difference between the groups (experience, confidence), and we have compared the development of the skills of the beginners with the experts. In case this is considered crucial by the reviewer or the editorial board, we are naturally happy to comply.

Page 8, Discussion, Is it possible to say something about sensitivity for CIN2+ before and after the course?

Please see also the answers to the previous questions on the same issue. The results show that the sensitivity to diagnose CIN 2+ lesion is not significantly improved after the course.  This issue is addressed in the discussion section page 6, lines 141-142:

“There was no improvement in the recognition of high versus low lesions and normal versus abnormal lesions after the course.”

Thank you for these references and thorough review of our work.

References

Desai M, Hadden P, Kitchener H, Martin-Hirsch P, Prendiville W, et al. (2010) Colposcopy and Programme Management. Guidelines for the NHS Cervical Screening Programme. NHSCSP Publication No 20 May.

Zuchna C, Hager M, Tringler B, Georgoulopoulos A, Ciresa-Koenig A, et al. (2010) Diagnostic accuracy of guided cervical biopsies: a prospective multicenter study comparing the histopathology of simultaneous biopsy and cone specimen. Am J Obstet Gynecol 203: 321–326.

Massad LS, Jeronimo J, Katki HA, Schiffman M (2009) The accuracy of colposcopic grading for detection of high-grade cervical intraepithelial neoplasia. J Low Genit Tract Dis 13: 137–144.

Pretorius RG, Belinson JL, Burchette RJ, Hu S, Zhang X, Qiao YL. Regardless of skill, performing more biopsies increases the sensitivity of colposcopy. J Low Genit Tract Dis. 2011 Jul;15(3):180-8. doi: 10.1097/LGT.0b013e3181fb4547. PMID: 21436729.

Roensbo MT, Hammer A, Blaakaer J. Can Dynamic Spectral Imaging System colposcopy replace conventional colposcopy in the detection of high-grade cervical lesions? Acta Obstet Gynecol Scand. 2015 Jul;94(7):781-785.

Underwood M, Arbyn M, Parry-Smith W, De Bellis-Ayres S, Todd R, Redman CW, Moss EL. Accuracy of colposcopy-directed punch biopsies: a systematic review and meta-analysis. BJOG. 2012 Oct;119(11):1293-301.

Sorbye SW, Arbyn M, Fismen S, Gutteberg TJ, Mortensen ES. HPV E6/E7 mRNA testing is more specific than cytology in post-colposcopy follow-up of women with negative cervical biopsy. PLoS One. 2011;6(10):e26022.

Round 2

Reviewer 2 Report

I recommended rejection for this manuscript

Author Response

Thank you for the thorough review. We have tried to address all comments and notes made by all reviewers. In case we could further improve the manuscript somehow, we are naturally happy to comply.